# Computed Tomography Texture Analysis of Carotid Plaque as Predictor of Unfavorable Outcome after Carotid Artery Stenting: A Preliminary Study

**DOI:** 10.3390/diagnostics11122214

**Published:** 2021-11-27

**Authors:** Davide Colombi, Flavio Cesare Bodini, Beatrice Rossi, Margherita Bossalini, Camilla Risoli, Nicola Morelli, Marcello Petrini, Nicola Sverzellati, Emanuele Michieletti

**Affiliations:** 1Radiology Unit, Department of Radiology, “Guglielmo da Saliceto” Hospital, Via Taverna 49, 29121 Piacenza, Italy; f.bodini@ausl.pc.it (F.C.B.); b.rossi@ausl.pc.it (B.R.); M.Bossalini2@ausl.pc.it (M.B.); c.risoli@ausl.pc.it (C.R.); n.morelli@ausl.pc.it (N.M.); m.petrini@ausl.pc.it (M.P.); E.Michieletti@ausl.pc.it (E.M.); 2Unit “Scienze Radiologiche”, Department of Medicine and Surgery (DiMeC), University of Parma, Via Gramsci 14, 43126 Parma, Italy; nicola.sverzellati@unipr.it

**Keywords:** carotid artery disorders, vascular accident, brain, computer software applications, stents

## Abstract

Novel biomarkers are advocated to manage carotid plaques. Therefore, we aimed to test the association between textural features of carotid plaque at computed tomography angiography (CTA) and unfavorable outcome after carotid artery stenting (CAS). Between January 2010 and January 2021, were selected 172 patients (median age, 77 years; 112/172, 65% men) who underwent CAS with CTA of the supra-aortic vessels performed within prior 6 months. Standard descriptors of the density histogram were derived by open-source software automated analysis obtained by CTA plaque segmentation. Multiple logistic regression analysis, receiver operating characteristic (ROC) curve analysis and the area under the ROC (AUC) were used to identify potential prognostic variables and to assess the model performance for predicting unfavorable outcome (periprocedural death or myocardial infarction and any ipsilateral acute neurological event). Unfavorable outcome occurred in 17/172 (10%) patients (median age, 79 years; 12/17, 70% men). Kurtosis was an independent predictor of unfavorable outcome (odds ratio, 0.79; confidence interval, 0.65–0.97; *p* = 0.029). The predictive model for unfavorable outcome including CTA textural features outperformed the model without textural features (AUC 0.789 vs. 0.695, *p* = 0.004). In patients with stenotic carotid plaque, kurtosis derived by CTA density histogram analysis is an independent predictor of unfavorable outcome after CAS.

## 1. Introduction

Stroke is the second cause of death in the developed world [1,2]. Carotid stenosis is responsible for almost 20% of strokes in the adult population and its treatment reduces stroke occurrence and related sequelae [3,4]. Carotid artery stenting (CAS) and carotid endarterectomy (CEA) are considered the main treatment of carotid plaque and showed similar rates of non-periprocedural ipsilateral stroke; however, the risk of subsequent stroke, particularly within the periprocedural time, is higher with CAS as compared to CEA [3,5]. Several features, including plaque vulnerability, are associated with stroke occurrence after CAS [6,7]. Age, complex carotid or aortic arch anatomy, intraplaque hemorrhage, lipid rich necrotic core, and rupture of the fibrous cap are risk factors for embolic complications after CAS, due to embolism of the intracranial arteries as a result of debris migration during the procedure [8]. Nevertheless, the treatment of carotid plaque is controversial and new potential biomarkers are required to determine the optimal type of treatment being either best medical therapy, CAS or CEA [9].

Generally, unstable plaques display at histopathology large necrotic core, thin fibrous cap, positive remodeling and microcalcifications [10]. Currently, visual assessment of the plaque is only partially successful for identifying vulnerable characteristics, limiting risk stratification [10]. For instance, in coronary artery disease, the napkin-ring sign (NRS) defined as a central area of low Hounsfield unit (HU) abutting the lumen surrounded by a ring-shaped hyperdense tissue, is associated with major adverse cardiac event [11]. Nevertheless NRS assessment at computed tomography angiography (CTA) based on visual qualitative process showed certain degree of inter-reader variability [10]. Morphological features of the carotid plaque related with cerebrovascular event, such as intraplaque hemorrhage, lipid-rich necrotic core, thin fibrous cap, neovascularization, and ulceration can be evaluated by CTA [12,13,14,15,16]. As for NRS, even visual assessment of carotid artery imaging is subjective and influenced by reviewer experience [17]. To overcome these limitations, new software-based techniques have been increasingly tested to extract additional information from medical imaging [18]. Texture analysis can transform in data simple and complex patterns at imaging [19]. In the coronary arteries, the radiomics-based machine learning analysis at CTA outperformed expert visual assessment in the identification of advanced atherosclerotic lesions [20]. Considering carotid plaque, both the wall volume and several characteristics of lipid clusters identified at CTA through software-based automatic segmentation are predictors of stroke [21]. Furthermore, textural analysis of the carotid plaque obtained by duplex ultrasound (DUS), magnetic resonance (MR), and CTA can identify vulnerable plaques associated with subsequent ischemic neurological event [22,23,24,25]. Moreover, detailed plaque texture analysis performed during DUS is able to predict the risk of perioperative neurological event or new ischemic brain lesions at MR after CEA [26]. Nevertheless, reports regarding imaging textural parameters of carotid plaques as predictors of unfavorable outcome after CAS are still lacking. Since CTA is less operator-dependent and carries fewer limitations than US for texture analysis, we designed this preliminary study to assess the potential role of carotid plaque texture analysis at CTA in identifying additional predictors of unfavorable outcome after CAS [27]. 

## 2. Materials and Methods

### 2.1. Study Population

This retrospective study was approved by the Local Ethics Committee (institutional review board -IRB- approval number 22/2021/OSS*/AUSLPC). The study included patients who underwent CAS between January 2010 and January 2021 for carotid bifurcation or internal carotid artery (ICA) stenosis as estimated by the North American Symptomatic Carotid Endarterectomy Trial (NASCET) method [28]. Indications for treatment with CAS followed the European Society of Cardiology (ESC) and European Society of Vascular Surgery (ESVS) guidelines [29,30,31,32]. Exclusion criteria were as follows: (1) absence of supra-aortic vessels CTA within 6 months before the procedure; (2) unsuccessful CAS; (3) incomplete clinical data; (4) unsuccessful software analysis of the carotid plaque segmentation at CTA. Demographics and clinical data were collected for all patients from electronic archives. In particular, carotid stenosis was considered symptomatic in case of any ipsilateral neurological ischemic event within six months before procedure. All patients underwent first-line imaging with DUS. In symptomatic patients, DUS was performed at the emergency department with acute neurological ischemic event and included Doppler velocity measurements for the evaluation of stenosis severity [30]. In asymptomatic patients, DUS of the carotid arteries was included within medical management of the stroke prevention to identify patients who will benefit from antiplatelet treatment [33]. Both symptomatic and asymptomatic patients eligible for CAS, were studied with CTA when DUS identified calcified or mixed calcified carotid plaque [30]. During the study period, 463 patients underwent CAS. In this case, 236 patients did not perform CTA of the supra-aortic vessels and were excluded. In the remaining patients CAS was unsuccessful in 19 patients for anatomical difficulties while clinical data were incomplete in 26 patients. For beam hardening artifacts, carotid plaque textural analysis was unreliable in 10 patients. Figure 1 shows the patients enrollment flowchart. Finally, 172 patients were included in the study, the majority were men (112/172, 65%) with a median age of 77 years-old (interquartile range, IQR: 70–82 years-old). Ipsilateral neurologic ischemic event occurred 6 months prior CAS in 90/172 (52%) patients. Hypertension, cardiac disease, and diabetes were identified, respectively, in 63% (109/172), 41% (71/172), and 30% (53/172) of the patients. Table 1 summarizes main demographics and clinical features of the study population, while complete data are showed in Appendix A.

### 2.2. CT Angiography of the Supra-Aortic Vessels Protocol

CT scans were obtained with either a 64-row CT scanner (Aquilon; Toshiba, Inc., Tokyo, Japan), or two different 16-row CT scanners (Emotion 16, Siemens AG, Forcheim, Germany; Brilliance 16, Philips Healthsystems, Amsterdam, Netherlands). Patients were scanned on supine position, from the aortic arch to the cranial vertex. Technical parameters for the 64-row scanner were as follows: 64 × 0.5 mm detector configuration, tube voltage 100 kV, tube current 120 mAs, and pitch 1.2. Technical parameters for the 16-row scanner were as follows: 16 × 0.6 mm or 16 × 1.5 mm detector configuration, tube voltage 110–120 kV, tube current 100–200 mAs, and pitch 0.9–1.3.

A variable amount (60–80 mL) of iodinated contrast medium (300 mgI/mL iohexol, Omnipaque 300; GE Healthcare, Little Chalfont, UK) was injected through a 18G cannula positioned in a superficial vein of the upper arm using an automatic dual-head injector (Medrad Stellant; Medrad, Palo Alto, PA, USA) with a flux of 3.5–4 mL/s followed by a 50-mL saline bolus. A bolus-tracking technique was used to trigger the acquisition 5 s after a threshold of 100 Hounsfield units (HU) was reached into the aortic arch. Image datasets were reconstructed axially using medium-soft kernel (FC 43 for Aquilon scanner, B-20 for Somatom 16, and standard filter B for Brilliance 16) at 1.5–2 mm slice thickness with 1–1.5 mm increment and completed by coronal and sagittal multiplanar reconstructions (MPR).

### 2.3. CAS Procedure

All patients were medicated with oral acetylsalicylic acid (ASA) 100 mg and clopidogrel 75 mg once a day for 5 days before the procedure. All CAS procedures were performed by an interventional radiologist as primary operator (E.M.) with 19 years of experience and >100 CAS documented procedures executed. After local anesthesia (lidocaine or xylocaine 2%), the common femoral artery (CFA) was percutaneously punctured and a 5Fr sheath was inserted. Following the intraoperative administration of sodium heparin 5000 UI, angiography of the aortic arch and of both supra-aortic and intracranial vessels was obtained. On the basis of the arch type, selective common carotid artery (CCA) angiogram was performed with different diagnostic catheter in all cases over a 0.035” Radifocus M Standard hydrophilic guidewire (Terumo, Tokyo, Japan). Later, the Terumo guidewire was exchanged with a 0.035” hydrophobic guidewire (Hi-Torque Supracore, Abbott Vascular, Santa Clara, CA, USA) and the diagnostic catheter substituted by an 8Fr (outer diameter) guiding catheter (Neuron Max, Penumbra Inc., Alameda, CA, USA) after changing the 5Fr introducer in the CFA with an 8Fr introducer. In pre-occlusive stenosis, a pre-dilation with a 2.5 × 20 mm (diameter × length) balloon was performed. Distal embolic protection device (EPD, Emboshield NAV6, Abbott Vascular) with different sizes (unconstrained filter diameter 5 mm or 7.2 mm for vessel diameter, respectively, included between 2.5–4.8 mm or 4–7 mm) was deployed over the plaque under roadmap. In each patient were deployed self-expanding closed-cell stent made of different material (Carotid WALLSTENT^TM^, Boston Scientific, Marlborough, MA, USA; XACT^TM^, Abbott Vascular). Stent post-dilatation was performed in all cases after atropine 0.5 mg i.v. administration and before removal of the distal EPD. A final angiogram was obtained to evaluate for residual stenosis and exclude vasospasm or dissection.

Procedure time from aortic arch angiogram to final check angiogram, was recorded in each case. A correct placement of the stent with a residual stenosis of less than 30% was defined as technical success. Periprocedural complications were categorized following Society of Interventional Radiology (SIR) clinical practice guidelines [34]. The antiplatelet therapy after procedure consisted in ASA 100 mg and clopidogrel 75 mg once a day for 2 months after the procedure, followed by lifelong oral ASA 100 mg once a day. All patients were administered with lipid-lowering therapy following current guidelines [35].

### 2.4. Imaging Assessment

All patients’ images (angiography and CTA) were anonymized and transferred to a dedicated workstation. Two radiologists, respectively, with 5 (D.C.) and 14 (F.C.B.) years of experience in vascular imaging, visually assessed aortic arch, supra-aortic vessels, and carotid plaques. Instances of discordance over the abovementioned features assessment were resolved by consensus. Arch type as defined by Madhwal et al., and the presence of bovine arch were analyzed in the aortic arch angiograms obtained during the procedure in the left anterior oblique (LAO) projection [36]. Calcification of the arch and the supra-aortic vessel was defined as clearly visible radiological densities within the vascular wall of the artery [37]. Angiographic characteristics of the plaques were assessed as follows: lesion length ≥15 mm considered from the proximal to the distal shoulder of the lesion in the projection that best elongated the stenosis (only the portion of stenosis that was ≥50% in symptomatic patients or ≥60% in asymptomatic patients was quantified); ulceration, when the plaque showed >2 craters of ≥3 mm in depth or with poorly defined edges and a hazy appearance; ostial location, if the maximal point of stenosis was located at the internal carotid artery ostium [7]. Angiographic plaque stenosis was assessed on the basis of the NASCET criteria [28]. CTA images were reviewed using dedicated window (window width, 600 HU; window level, 170 HU). At CTA carotid plaque were categorized as non-calcified plaque (NCP), totally calcified plaque (CP), or mixed plaque (MP); the hypodense component of the plaque was considered heterogeneous if at least two regions of different attenuation could be visually distinguished; otherwise, it was classified as homogenous [38]. In case of heterogenous hypodense component of the plaque, the presence of a central area of low CT attenuation in contact with the lumen surrounded by higher attenuation was defined as napkin-ring sign (NRS) [38,39]. Relapsing plaque after surgical endarterectomy were considered “recurrent plaque”.

The software-based evaluation of the carotid plaques was performed by the Radiomics extension (Computational Imaging and Bioinformatics Lab, Harvard Medical School; Boston, MA, USA) of the open-source 3D Slicer software (version 4.10.2, https://www.slicer.org, accessed on 20 November 2021) [40]. A radiologist (D.C.) and a radiology technician (C.R.) both with 5 years of experience independently accomplished manual segmentation at CTA of the carotid plaque whole volume. The time to obtain the segmentations was recorded in each patient. The segmentation process was performed on original CTA images (Figure 2), without normalization or resegmentation, since it does not seem necessary at CT, considering that grey values are calibrated to HU [41]. Resampling with linear interpolation was used for obtaining isotropic images. Fixed bin width (BW) at 25 was chosen before textural feature extraction. After the first-order analysis the following standard descriptors of the density histogram were recorded for each plaque: mean, standard deviation, kurtosis, and skewness. Mean corresponded to the average gray level intensity within the segmentation; standard deviation measures the amount of variation or dispersion in gray level intensity from the mean; kurtosis is a measure of the ‘peakedness’ of the distribution of values in the segmentation, with a lower kurtosis meaning that the mass of the distribution is concentrated towards mean rather than the tail(s); skewness measures the asymmetry of the distribution of density values around the mean value. The whole plaque volume was also registered.

### 2.5. End-Point Definition

The occurrence of any acute neurological event (transient ischemic attack, TIA; minor stroke; major stroke; fatal stroke) ipsilateral to the procedure was evaluated by a neurologist after CAS. An acute neurological event with focal symptoms or signs consistent with focal cerebral ischemia lasting for less than 24 h was considered “TIA”, while “Stroke” if persisted for 24 h or more [7,42]. “Stroke” were also classified as “major stroke” if the National Institutes of Health Stroke Scale (NIHSS) score was ≥9 points after 90 days, or, otherwise, as “minor stroke” [42]. Death for an ischemic or intracerebral hemorrhagic stroke was considered “fatal stroke”. Neurologic evaluation was performed at baseline, the procedure day, at hospital dismission, one month afterward, and every six months thereafter. The evaluation consisted of the use of the NIHSS scale and the TIA-Stroke Questionnaire [43,44]. Creatine kinase MB or troponin level that was twice the upper limit of the normal range or higher associated with chest pain or symptoms consistent with ischemia or electrocardiographic evidence of ischemia were defined as myocardial infarction [45]. The primary end point of the study was a composite of any ipsilateral stroke (within or after 30 days following CAS) and myocardial infarction or any death during the periprocedural period (within 30 days after CAS). The time elapsed from the procedure to the occurrence of the primary end-point was also recorded.

### 2.6. Statistical Analysis 

Categorical variables were expressed as counts and percentage. Continuous variables are shown as median and interquartile range (IQR). The difference between patients with and without unfavorable outcome occurrence were assessed by Mann-Whitney U test for continuous variables and Chi-square test or Fisher’s exact test for categorical variables, as appropriate. The Wilcoxon test and the intra-class correlation coefficient (ICC) were used, respectively, to evaluate differences in plaque segmentation time between readers and to test the inter-rater agreement for the carotid segmentation plaque histogram-based parameters; the interpretation of ICC was based on the guidelines provided by Koo and Li [46].

Multiple logistic regression analysis was used to examine the association between potential prognostic variables and unfavorable outcome to estimate odd ratios (OR) and 95% confidence interval (CI). Variables were evaluated separately in univariable analysis. The multivariable analysis was performed using the stepwise method on the basis of the Akaike information criterion. Therefore, were obtained two models for predicting unfavorable outcome: (1) including demographics, comorbidities, anatomical features, main visual plaque characteristics, and procedure duration; (2) including descriptors of the density histogram (mean, standard deviation, kurtosis, and skewness) in addition to variables included in the previous model. Receiver operating characteristics (ROC) curve analysis was performed for each model and the area under the ROC (AUC) was used to assess the performance of the discrimination models based on independent predictors. The ROC curves of the models were compared by the methodology of DeLong et al. [47]. A *p*-value < 0.05 was considered significant. Statistical analysis was performed using MedCalc software (version 14.8.1, MedCalc Software Ltd., Ostend, Belgium).

## 3. Results

### 3.1. Patients Demographics, Clinical, Anatomical, and Procedural Findings

Demographics, clinical, anatomical, and procedural features are summarized in Table 1 and showed in details in Appendix A. Unfavorable outcome occurred in 17/172 (10%) of the patients included in the study. In the periprocedural period occurred 7/17 (41%) TIA or minor stroke, 1/17 (6%) major stroke, 1/17 (6%) myocardial infarction, and 2/17 (12%) death. In the post-procedural time occurred 1/17 (6%) TIA, and 5/17 (29%) major stroke. No significant differences were found between patients with unfavorable outcome and the remaining patients for gender (female rate 30% vs. 35%, *p* = 0.790) and age (median age 79 vs. 77 years-old, *p* = 0.243). The rate of symptomatic patients was similar in the two groups (41% for unfavorable outcome vs. 53% for the remaining patients, *p* = 0.444). Patients with unfavorable outcome showed similar rate of cardiac disease and diabetes, occurring respectively in 64% (vs 38%, *p* = 0.066) and in 35% (vs 30%, *p* = 0.782) of the cases. No significant differences were found for aortic arch type II-III (17% for unfavorable outcome vs. 40% for the remaining patients, *p* = 0.071) and bovine arch (47% for unfavorable outcome vs. 27% for the remaining patients, *p* = 0.158). Median procedural time was similar (*p* = 0.098) between patients with unfavorable outcome (15 min) and the remnants (18 min).

### 3.2. Plaque Visual Assessment and Texture Parameters

The main description of both visual and textural plaque assessment is summarized in Table 2 and detailed in Appendix A. A significant higher rate of ulcerated plaque was detected in patients with unfavorable outcome as compared to the remaining patients (58% vs. 30%, *p* = 0.029). No significant differences amongst patients with unfavorable outcome and the remnants were found regarding side (right side 64% vs. 52%, *p* = 0.443) non-calcified plaque rates (6% vs. 3%, *p* = 0.469), NRS (17% vs. 9%, *p* = 0.765), ostial location (47% vs. 53%, *p* = 0.620), plaque length ≥15 mm (29% vs. 51%, *p* = 0.123), recurrent plaque (11% vs. 10%, *p* = 0.693), and median percentage of carotid lumen stenosis (64% vs. 69%, *p* = 0.232).

The median time required for carotid plaque segmentation was significantly (*p* < 0.0001) longer for the technician (193 s, IQR 146–255 s) as compared to the radiologist (154 s, IQR 140–180 s). The inter-reader agreement for software-based findings was excellent for all parameters. The ICC value was 0.924 (95%CI 0.894–0.946) for plaque volume, 0.959 (95%CI 0.941–0.971) for the mean, 0.980 (95%CI 0.972–0.986) for standard deviation, 0.931 (95%CI 0.903–0.951) for skewness, and 0.939 (95%CI 0.914–0.956) for kurtosis. Median kurtosis was significantly lower in patients with unfavorable as compared to patients with good outcome (5.37 vs. 5.84, *p* = 0.048). No significant differences between the two groups were identified for median plaque volume (unfavorable outcome vs. good outcome, 82 cc vs. 114 cc; *p* = 0.467), mean density (244 HU vs. 222 HU, *p* = 0.570), standard deviation (254 HU vs. 228 HU, *p* = 0.713), and skewness (1.53 vs. 1.67, *p* = 0.093).

### 3.3. Predictive Analysis

Logistic regression analysis is showed in Appendix A. At univariable analysis cardiac disease (OR 2.91; 95% CI 1.01–8.26; *p* = 0.045), plaque ulceration (OR 3.18; 95% CI 1.14–8.86; *p* = 0.026), and kurtosis (OR 0.82; 95% CI 0.68–0.99; *p* = 0.043) were significantly associated with unfavorable outcome. Table 3 summarizes multivariable logistic regression analysis results. At multivariable analysis without textural features, cardiac disease (OR 3; 95% CI 1.03–8.71; *p* = 0.042), and plaque ulceration (OR 3.28; 95% CI 1.16–9.31; *p* = 0.025) were significant predictors of unfavorable outcome after CAS. Including textural features at the previous model, cardiac disease (OR 3.05; 95% CI 1.02–9.09; *p* = 0.045), and plaque ulceration (OR 3.96; 95% CI 1.34–11.72; *p* = 0.012) were confirmed as significant predictors of unfavorable outcome, with the addition of kurtosis (OR 0.79; 95% CI 0.65–0.97; *p* = 0.029). A significant (*p* = 0.004) higher AUC (Figure 3) was observed for the model with textural features (0.789, 95% CI 0.73–0.847) as compared to the model without textural features (0.695, 95% CI 0.62–0.763).

## 4. Discussion

In the present study we aim to test the association between textural parameters obtained by open-source software on CTA of the carotid plaque and unfavorable outcome after CAS, defined as myocardial infarction or death within 30 days after CAS and any ipsilateral neurologic acute event (within and after 30 days following CAS). Kurtosis of the carotid plaque density histogram was the only textural parameter identified as independent predictor of unfavorable outcome after CAS. To our knowledge, this is the first study demonstrating the association between carotid plaque textural features at CTA and unfavorable outcome after CAS. Currently, in patients who require carotid plaque revascularization, the ESC/ESVS guidelines suggest CEA or CAS depending on age, presence of symptoms in previous six months, grade of stenosis, and perioperative risk of stroke or death [31]. On the basis of previous randomized controlled trials (RCTs), procedural safety and long-term efficacy in preventing recurrent stroke favors CEA over CAS in symptomatic patients for the extra risk of periprocedural minor stroke in patients older than 70 years-old; nevertheless superiority of CEA over CAS has not been yet demonstrated for remaining patients [48]. Future directions in imaging of carotid atherosclerosis aims to guide therapeutical approaches and improve long-term outcomes. Therefore accurate patient risk stratification regarding periprocedural stroke is important to support individual revascularization procedure (CEA or CAS). Amongst plaque CTA features assessed visually, intraplaque hemorrhage, lipid-rich necrotic core, thin fibrous cap, neovascularization, and ulceration are risk factors for cerebrovascular event [12,13,14,15,16]. Nevertheless evaluation of carotid plaque soft component is the major limit of CTA visual assessment, with low sensitivity particularly for intraplaque hemorrhage and small plaques [14]. Despite plaque ulceration, lesion length ≥15 mm, and ostial location are considered predictors of periprocedural stroke after CAS, current guidelines consider only perioperative risk of death or ipsilateral stroke in addition to symptoms and grade of stenosis to decide whether to perform CEA or CAS, without consider plaque characteristic [7,31]. For these reasons novel objective biomarker are required to assess patient risk for selecting the more appropriate treatment.

In agreement with other studies, in the present series plaque ulceration is related to unfavorable outcome after CAS [7]. Ulceration of carotid plaque is widely considered a marker of vulnerability [49]. In particular the presence of ulceration in carotid plaque that underwent CAS increased the risk to release small emboli during the manipulation of the carotid vessels [7]. 

In addition to concomitant cardiac disease, and plaque ulceration, we demonstrated that kurtosis of the density histogram is an independent predictor of unfavorable outcome after CAS; furthermore the predictive model including kurtosis outperformed the traditional model based on demographics, comorbidities, anatomical and plaque features assessed visually for the prediction of unfavorable outcome after CAS. Many authors investigated the association of carotid plaque textural parameters and neurological acute event or plaque histology [22,23,24,25,26,50]. According to the present study, other reports based on radiomic features derived from MR of both carotid and basilar arteries showed high performance of the predicting models for future neurovascular event which included textural parameters [23,24]. Zaccagna et al. reported that higher histogram-derived skewness of carotid plaque at CTA representing micro-vessel proliferation or ulceration, is a predictor of subsequent TIA or stroke during a mean follow-up of around 23 months [22]. In the present report, we failed to identify a significant association between skewness and unfavorable outcome after CAS. This disagreement might be explained by several reasons: (1) in contrast to Zaccagna et al., we tested the association of textural parameters with unfavorable outcome adjusted by other variables, such as visual assessment of ulceration which is the visual equivalent of skewness, likely more powerful and retained in the final model; (2) we tested different outcome, including any acute neurological event and periprocedural death or myocardial infarction after CAS instead of future neurological event in patients with carotid atherosclerosis [22]. In our study lower kurtosis values were associated with higher risk of unfavorable outcome after CAS. Lower kurtosis means lighter tails, and more uniformity of the plaque density [51]. This finding supports the results obtained by Doonan et al., that reported textural plaque homogeneity at DUS as predictor of instability at histology, associated with larger lipid core, greater degree of plaque inflammation, and less fibrous tissue [52]. In literature, only one report tested the association of textural parameters in predicting outcome after carotid plaque revascularization [26]. Namely, Madycki et al. showed that hyperechogenic plaque quantified by software at DUS were associated with less probability of brain microembolism after CEA [26]. To our knowledge, the present study is the first that demonstrated an association between unfavorable outcome and CTA textural features of the carotid plaque in patients who underwent CAS. Despite the recent update of the ESVS regarding the management of patient with carotid stenosis <70%, the treatment for asymptomatic patients with stenosis >60% and symptomatic patients with stenosis 50–69% is still controversial due to conflicting evidence [22,31]. Thus, the potential ability of CTA textural analysis in predicting unfavorable outcome after CAS could be helpful to better stratify patient at risk to develop post-procedural cerebrovascular events and consequently to decide which revascularization option between CEA and CAS is better for the patient. As compared to DUS, CTA is more reliable, less operator-dependent, and more suitable for obtaining quantitative imaging biomarker. Furthermore, it has been demonstrated that texture analysis at DUS is affected by large variability depending on cardiac phase of image acquisition, plaque size and echogenicity, determining a reclassification of the plaque in 16–25% of the cases [27]. As compared to MR, CTA is more available, and the segmentation process is faster since it can be accomplished on only one contrast-enhanced phase, opposite to the multiple sequence required in MR. In the present study the segmentation process was accomplished in around 3 min from both radiologist or technician, with excellent inter-observer agreement, demonstrating its feasibility even during routine daily work-up.

The present study has several limitations. First, only 37% of the patients underwent CAS in the study period were retained in our sample, resulting in a representative but small cohort of patients. Second, the absolute number of post-procedural unfavorable outcome was also relatively small. Third, CTA were performed with different scanners, potentially affecting the texture analysis. Fourth, as preliminary study, the models proposed were based only on training dataset with possible model overfitting; further studies with both validation and test datasets are required. Fifth, the composite outcome evaluated is different from the standard outcome of any stroke\death within 30 days after the CAS.

## 5. Conclusions

In conclusion, we aimed to test if textural parameters of the carotid plaque at CTA are predictors of unfavorable outcome after CAS. Lower kurtosis was associated with higher risk of any neurological acute event and periprocedural myocardial infarction or death after the intervention. Our methods, based on textural analysis of the carotid plaque at CTA, is feasible by radiologist or technician in few minutes and superior to both DUS (less dependent from operator, plaque characteristics or cardiac cycle) and MR (higher CT scanner availability with faster segmentation process performed on only one set of images). Furthermore, textural analysis of the carotid plaque could provide additional objective biomarker to steer therapeutic decision, considering that current trends suggest not only to consider luminal stenosis to decide whether or not to treat carotid plaque and with which option (CEA, CAS or best medical therapy). Future perspective studies with higher number of patients are required with validation and test datasets to confirm our findings.

## Figures and Tables

**Figure 1 diagnostics-11-02214-f001:**
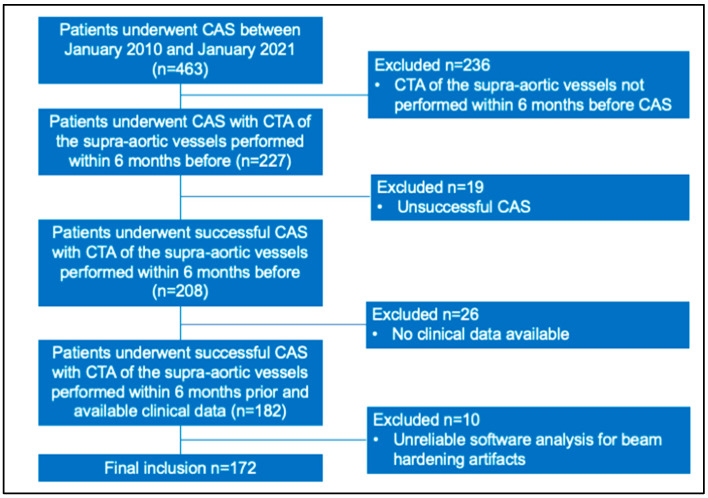
The diagram shows the patient enrollment flowchart. Abbreviations: CAS, carotid artery stenting; CTA, computed tomography angiography.

**Figure 2 diagnostics-11-02214-f002:**
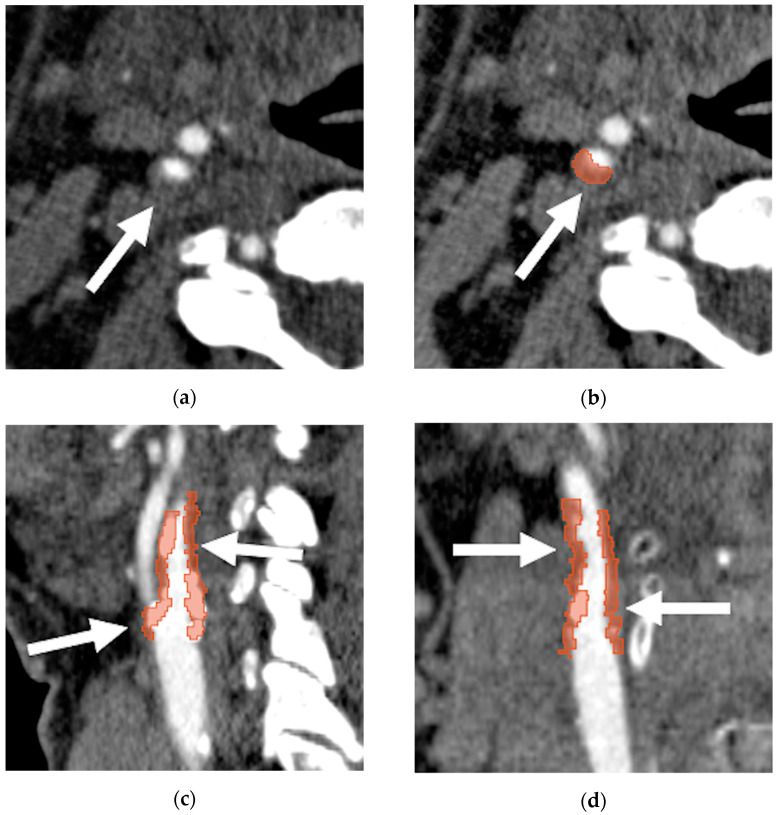
Symptomatic mixed plaque of the right carotid bifurcation extended to the proximal internal carotid artery: (**a**) the axial computed tomography angiography image shows the internal carotid artery plaque (arrow) that determined 70% lumen stenosis, following North American Symptomatic Carotid Endarterectomy Trial (NASCET) criteria [28]; the whole plaque was segmented manually with the open-source 3D Slicer software (https://www.slicer.org/, accessed on 1 October 2021) obtaining volumetric ROI as showed in (**b**) axial, (**c**) sagittal, and (**d**) coronal multiplanar reformatted reconstruction derived from computed tomography angiography (arrows).

**Figure 3 diagnostics-11-02214-f003:**
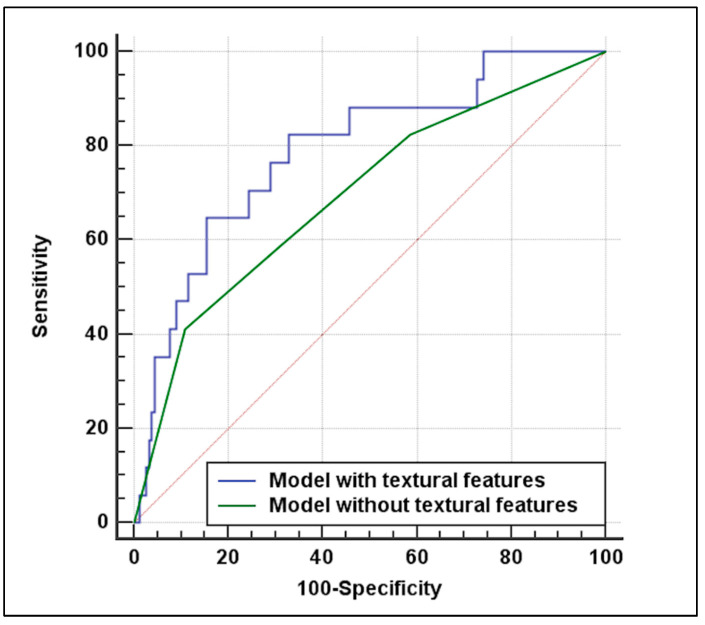
Diagnostic performance for prediction of unfavorable outcome (any ipsilateral neurologic acute event and periprocedural myocardial infarction or death) after carotid artery stenting. The model including textural features derived by density histogram of the plaque at CTA (blue line) outperformed the model without textural features (green line) with an area under the receiver operating characteristics curve of 0.789 vs. 0.695 (*p* = 0.004).

**Table 1 diagnostics-11-02214-t001:** Main demographics, clinical, anatomical, and procedural carotid artery stenting findings.

Variables	Overall (*n* = 172)	Good Outcome (*n* = 155)	Unfavorable Outcome (*n* = 17)	*p*-Value
Gender male females	112 (65%) 60 (35%)	100 (65%) 55 (35%)	12 (70%) 5 (30%)	0.790
Age	77 (70–82)	77 (70–82)	79 (72–83)	0.243
Ipsilateral neurological ischemic event within 6 months before CAS	90 (52%)	83 (53%)	7 (41%)	0.444
Cardiac disease	71 (41%)	60 (38%)	11 (64%)	0.066
Diabetes	53 (30%)	47 (30%)	6 (35%)	0.782
Arch type II-III	66 (38%)	63 (40%)	3 (17%)	0.071
Bovine arch	51 (30%)	43 (27%)	8 (47%)	0.158
Arch calcifications	131 (76%)	115 (74%)	16 (94%)	0.076
Procedural time (min)	18 (15–22)	18 (16–22)	15 (15–20)	0.098

Categorical data are showed as number and percentage in parenthesis. Continuous variables are presented as median with interquartile range in parenthesis. Abbreviations: CAS, carotid artery stenting.

**Table 2 diagnostics-11-02214-t002:** Main CTA features, plaque visual assessment and texture parameters.

Variables	Overall (*n* = 172)	Good Outcome (*n* = 155)	Unfavorable Outcome (*n* = 17)	*p*-Value
Side right left	92 (53%) 80 (47%)	81 (52%) 74 (48%)	11 (64%) 6 (36%)	0.443
Visual plaque classification mixed non-calcified	166 (96%) 6 (4%)	150 (97%) 5 (3%)	16 (94%) 1 (6%)	0.469
Plaque ulceration	58 (33%)	48 (30%)	10 (58%)	0.029
Ostial plaque	91 (52%)	83 (53%)	8 (47%)	0.620
Angiographic stenosis (%)	68 (60–75)	69 (60–75)	64 (51–74)	0.232
Plaque length ≥15 mm	85 (49%)	80 (51%)	5 (29%)	0.123
Recurrent plaque	18 (10%,)	16 (10%)	2 (11%)	0.693
Plaque mean density (HU)	225 (146–353)	222 (144–349)	244 (158–389)	0.570
Plaque standard deviation density (HU)	229 (142–340)	228 (141–336)	254 (146–365)	0.713
Plaque kurtosis	5.75 (3.91–9.31)	5.84 (3.96–9.97)	5.37 (3.27–6.32)	0.048
Plaque skewness	1.63 (1.13–2.26)	1.67 (1.14–2.36)	1.53 (0.96–1.68)	0.093

Categorical data are showed as number and percentage in parenthesis. Continuous variables are presented as median with interquartile range in parenthesis. Angiographic stenosis assessment was based on North American Symptomatic Carotid Endarterectomy Trial (NASCET) criteria [28]. Abbreviations: CTA, computed tomography angiography; HU, Hounsfield units.

**Table 3 diagnostics-11-02214-t003:** Multivariable logistic regression analysis for the relationship between clinical, anatomical and textural plaque features to predict unfavorable outcome after carotid artery stenting (*n* = 172).

Variables			
*Multivariable analysis without textural features*			
	Coefficient	OR (95%CI)	*p*-value
Cardiac disease	1.09	3 (1.03–8.71)	0.042
Plaque ulceration	1.19	3.28 (1.16–9.31)	0.025
*Multivariable analysis with textural features*			
Cardiac disease	1.11	3.05 (1.02–9.09)	0.045
Plaque ulceration	1.37	3.96 (1.34–11.72)	0.012
Plaque kurtosis	−0.22	0.79 (0.65–0.97)	0.029

Abbreviations: CI, confidence interval; OR, odds ratio.

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
