# Peer review of "Computed Tomography Texture Analysis of Carotid Plaque as Predictor of Unfavorable Outcome after Carotid Artery Stenting: A Preliminary Study"

_diagnostics, 2021, doi:10.3390/diagnostics11122214_

Round 1

Reviewer 1 Report

Dear authors:

1- Please just report one result in the table 1. The current presentation of results is confusing. You can move the most important results in the text or supplementary materials.

2-Also apply the comment 1 to table 2. You can also think of a better way to report the results. Range is not a good indicator.

3- For the logistic regression you can choose stepwise logistic regression to select the most important factors or manually insert and remove the factors to find the most important contributors. The present logistic regression consists of many IV and might be confusing to interpret.

Author Response

To Reviewer 1

Dear Reviewer 1,

We thank you for the time you dedicate to read our manuscript and for your suggestions and comments. We believe they have helped to make our paper stronger, especially in clarifying some points that were lacking or equivocal. According to your recommendations, we have revised our manuscript as follows (changes in text are in red). In particular we have summarized Tables and the Results section as suggested, moving details in Supplementary Materials; in addition we have provided a multivariable logistic regression analysis with the stepwise method, showing in Table 3 multivariable analysis and moving all the analysis in Supplementary Materials.  Accompanying this letter, please find a marked-up copy of the changes made from the previous article (“Track-changed version” file). Detailed responses for to the reviewers are included down below.

Comment 1: please just report one result in the table 1. The current presentation of results is confusing. You can move the most important results in the text or supplementary materials.

Authors response: thank you for this important remark. We have amended Table 1 as suggested by deleting the 95%CI of the percentage and by reducing the number of variables reported. We have also summarized the results in the text (see comment 2), moving the complete results in Table S1 of the Supplementary Materials.

Comment 2: also apply the comment 1 to table 2. You can also think of a better way to report the results. Range is not a good indicator.

Authors response: thank you for this suggestion. As in comment 1 response, we have amended Table 2 as suggested by deleting the 95%CI of the percentage and by reducing the number of variables reported. As stated in “Statistical Analysis” section of methods in all tables and throughout the text we have reported continuous variables as median and interquartile range (IQR), not with range.

Comment 3: for the logistic regression you can choose stepwise logistic regression to select the most important factors or manually insert and remove the factors to find the most important contributors. The present logistic regression consists of many IV and might be confusing to interpret.

Authors response: thank you for this main remark. We have changed multivariable logistic regression analysis method, by using the stepwise method as suggested. As explained in “Statistical Analysis” section of methods, two models were obtained, first without textural features and then with textural features, comparing the AUC of the two models. The aim was to test if textural features significantly improve the model based only on clinical-demographics, anatomical characteristics, and visual plaque features variables. We have changed Table 3 accordingly, presenting only variables significant at multivariable analysis of the two models, moving univariable analysis in Table S3 of the Supplementary Materials. We have also changed the ROC curves in Figure 3, based on the models obtained with the stepwise method, nevertheless the model with textural features significantly outperformed the model only based on clinical-demographics, anatomical characteristics, and visual plaque features variables.

Sincerely,

Dr. Davide Colombi, MD for the Authors

Reviewer 2 Report

Work described with care and attention in all its parts. Excellently conducted. Innovative. Limitations are present, however they are described, explained and clarified. It is a work destined to have resonance in the scientific world since there are many unresolved questions regarding the recruitment and the indications for the treatment of CEA or instead of CAS. 

Author Response

To Reviewer 2
Comment 1: work described with care and attention in all its parts. Excellently conducted. Innovative. Limitations are
present, however they are described, explained and clarified. It is a work destined to have resonance in the scientific
world since there are many unresolved questions regarding the recruitment and the indications for the treatment of CEA
or instead of CAS.
Authors response: thank you for reviewing our paper. We are delighted about your comments about the present
manuscript. 

Round 2

Reviewer 1 Report

Thank you very much